# Patterns of island change and persistence offer alternate adaptation pathways for atoll nations

Paul S. Kench [1], Murray R. Ford[1] & Susan D. Owen[1]

Sea-level rise and climatic change threaten the existence of atoll nations. Inundation and erosion are expected to render islands uninhabitable over the next century, forcing human migration. Here we present analysis of shoreline change in all 101 islands in the Pacific atoll nation of Tuvalu. Using remotely sensed data, change is analysed over the past four decades, a period when local sea level has risen at twice the global average (~$3.90 \pm 0.4$ mm.yr$^{-1}$). Results highlight a net increase in land area in Tuvalu of 73.5 ha (2.9%), despite sea-level rise, and land area increase in eight of nine atolls. Island change has lacked uniformity with 74% increasing and 27% decreasing in size. Results challenge perceptions of island loss, showing islands are dynamic features that will persist as sites for habitation over the next century, presenting alternate opportunities for adaptation that embrace the heterogeneity of island types and their dynamics.

[1] School of Environment, University of Auckland, Private Bag, 92010 Auckland, New Zealand. Correspondence and requests for materials should be addressed to P.S.K. (email: p.kench@auckland.ac.nz)

Understanding of human migration patterns and population relocation through the Pacific, since earliest settlement, has been informed by insights into the geologic template of atoll island formation and the influence of environmental change (including sea level) in modulating the habitability of islands[1,2]. Consequently, islands have been conceptualised as pedestals for human occupation, presenting opportunities for resource development and settlement, with their formation critical in the migration of peoples through the Pacific[1]. Questions of contemporary, and near future, atoll island habitability and persistence are equally framed against a backdrop of environmental change, and in particular, climate-driven increases in sea level[3,4].

Climate change remains one of the single greatest environmental threats to the livelihood and well-being of the peoples of the Pacific[5]. The fate of small island states confronted with the spectre of sea-level rise has raised global concern, and prompted a labyrinth of international programmes to consider how Pacific nations can and should adapt to the threats of climatic change[6]. Islands considered most at risk of physical destabilisation are low-lying atoll nations[7,8]. Erosion, combined with increased frequency of overwash flooding of island margins[4] is expected to render islands uninhabitable[9,10]. Incremental and event-driven climatic changes to ecological systems also present additional future stresses for island habitability, including the tolerance of agriculture crops to increased soil salinity, as well as concerns about water security, both in the context of drought and salt water intrusion of groundwater[11–13].

Under these environmental scenarios, conjectures of habitability and mobility become entwined and have driven an urgency in socio-political discourse about atoll nation futures and human security[14,15]. Strategies for adaptation to changing biophysical conditions are coupled with narratives of environmentally determined exodus[16]. Such persistent messages have normalised island loss and undermined robust and sustainable adaptive planning in small island nations[17]. In their place are adaptive responses characterised by in-place solutions, seeking to defend the line and include solutions such as reclamation and seawalls[18,19], potentially reinforcing maladaptive practices. Notwithstanding the maladaptive outcomes of such approaches[15,20] such dialogues present a binary of stay and defend the line or eventual displacement. There is limited space within these constructs to reflect on possibilities that a heterogeneous archipelago (size, number and dynamics of islands) may offer in terms of sustained habitability, beyond the historic imprint of colonial agendas and entrenched land tenure systems that may constrain novel adaptation responses at the national scale[7,21,22].

Amid this dispiriting and forlorn consensus, recent commentators have queried whether the loss of islands can be avoided and ask whether a more optimistic prognosis exists for atoll nations[17]. We argue that indeed there are a more nuanced set of options to be explored to support adaptation in atoll states. Existing paradigms are based on flawed assumptions that islands are static landforms, which will simply drown as the sea level rises[4,23]. There is growing evidence that islands are geologically dynamic features that will adjust to changing sea level and climatic conditions[24–27]. However, such studies have typically examined a limited number of islands within atoll nations, and not provided forward trajectories of land availability, thereby limiting the findings for broader adaptation considerations[17]. Furthermore, the existing range of adaptive solutions are narrowly constrained and do not reflect the inherent physical heterogeneity and dynamics of archipelagic systems.

Here we present the first comprehensive national-scale analysis of the transformation in physical land resources of the Pacific atoll nation Tuvalu, situated in the central western Pacific (Supplementary Note 1). Comprising 9 atolls and 101 individual reef islands, the nation is home to 10,600 people, 50% of whom are located on the urban island of Fogafale, in Funafuti atoll[28]. We specifically examine spatial differences in island behaviour, of all 101 islands in Tuvalu, over the past four decades (1971–2014), a period in which local sea level has risen at twice the global average (Supplementary Note 2). Surprisingly, we show that all islands have changed and that the dominant mode of change has been island expansion, which has increased the land area of the nation. Results are used to project future landform availability and consider opportunities for a vastly more nuanced and creative set of adaptation pathways for atoll nations.

## Results

**Planform island change.** Analysis of atoll island change aggregated across Tuvalu reveals three striking features of island areal transformation over the past four decades (Table 1, Fig. 1, Supplementary Data 1). First, only one island has been entirely eroded from the data set of 101 islands. This island had an initial size of 0.08 ha and was located on the reef rim of Nukufetau atoll. Second, total land area of the nation has expanded by 73.5 ha (2.9%) since 1971. Notably, eight of nine atolls experienced an increase in land area. Nanumea was the only atoll where a loss in land was detected, although this totalled less than 0.01%. Third, there are marked differences in the magnitude and direction of areal change between islands. A total of 73 islands (of 101) had a

### Table 1 Summary of atoll island characteristics and changes in islands, Tuvalu

| Atoll/Reef platform (RP) | No islds. | Atoll land area (ha) | Change in land area 1971–2014 (ha) | Change in land area 1971–2014 (%) | Number of islands Accr. | Number of islands Erod. | Inhabited islands No. | Inhabited islands Area (km²) | Inhabited islands Pop. density (per km²) |
|---|---|---|---|---|---|---|---|---|---|
| Nanumea | 6 | 356.1 | −1.32 | −0.004 | 3 | 3 | 1 | 2.18 | 281 |
| Niutao (RP) | 1 | 235.2 | 0.34 | 0.14 | 1 | — | 1 | 2.35 | 295 |
| Nanumaga (RP) | 1 | 301.0 | 4.71 | 1.56 | 1 | — | 1 | 3.01 | 183 |
| Nui | 13 | 342.8 | 7.61 | 2.22 | 13 | — | 1 | 1.34 | 544 |
| Vaitupu (RP) | 8 | 522.9 | 12.27 | 2.35 | 6 | 2 | 2 | 5.18 | 297 |
| Nukufetau | 26 | 314.4 | 19.40 | 6.17 | 15 | 11 | 1 | 0.19 | 3458 |
| Funafuti | 29 | 261.2 | 10.06 | 3.85 | 19 | 10 | 1 | 1.59 | 3427 |
| Nukulaelae | 19 | 176.4 | 10.00 | 5.67 | 16 | 3 | 1 | 0.22 | 1626 |
| Niulakita (RP) | 1 | 42.1 | 0.05 | 0.12 | 1 | — | 1 | 0.42 | 109 |

Island population data obtained from the Tuvalu Census of Population and Housing[28]
*Accr.* accreted islands, *Erod.* eroded islands

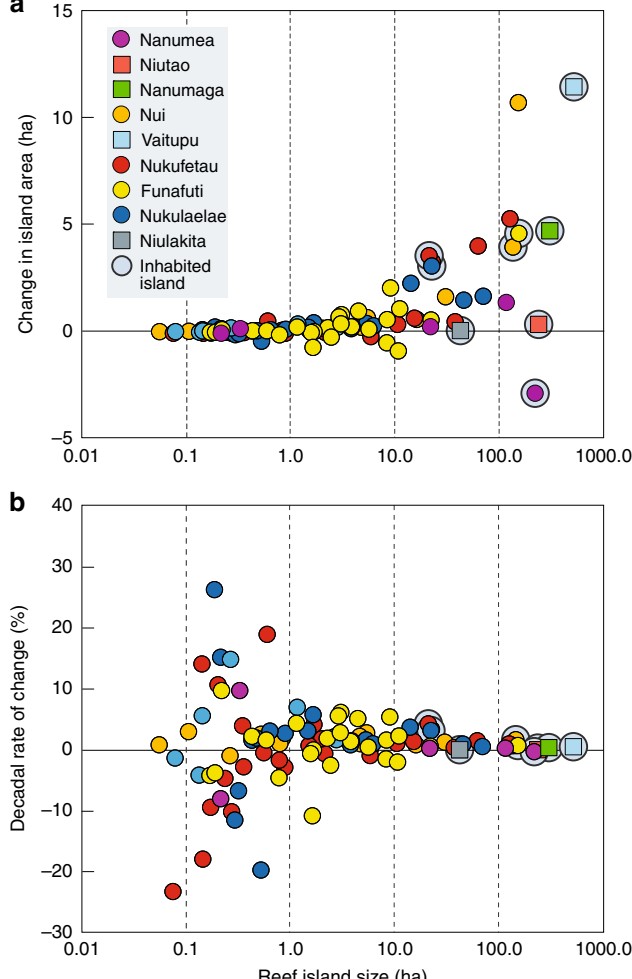

**Fig. 1** Summary data of physical island change of islands in Tuvalu between 1971 and 2014. **a** Absolute changes in island area in hectares with respect to island size. **b** Percentage change in islands per decade with respect to island size. Raw data contained in Supplementary Data 1. Note: square symbols denote reef platform islands; solid circles denote atoll rim islands; and light blue circles enclosing symbols denote populated islands

net increase in area, totalling 80.7 ha, with a range from <1 to 113% growth. These expanding islands had an average increase in area of 2.18 ha. Largest absolute increases in island area occurred on the reef platform islands of Vaitupu (11.4 ha, 2.2%) and Nanumaga (4.7 ha, 1.6%), and the atoll rim islands of Nui (10.4 ha, 7.1%), Nukufetau (5.3 ha, 4.2%) and the capital island of Funafuti atoll, Fogafale (4.6 ha, 3%). The remaining 28 islands (27.7% of total) decreased in area, totalling −7.24 ha and ranging from 1 to 100% reduction. On average, eroding islands decreased in area by −0.5 ha (22.69%). Of note, erosion was most prevalent on the smallest islands in the archipelago. Four islands decreased in area by more than 50%, although these were all islands that had an initial size of less than 0.5 ha. Largest absolute decreases in island area occurred on reef rim islands in Nanumea (−2.88 ha, 1.32%), and three islands on the western rim of Funafuti atoll, Tepuka (−0.89 ha, 8.35%), Fuagea (−0.74 ha, 45.5%) and Fualifeke (−0.51 ha, 6.16%).

**Shoreline dynamics**. Analysis of shoreline dynamics at the transect scale highlights substantive site-specific changes around island shorelines (Fig. 2). Of the 19,403 shoreline transects analysed, 44% (8583) exhibited accretion, 33% (6338) remained stable and 23% (4482) showed evidence of erosion over the analysis period. Notably, on the vast majority of islands both erosion and accretion were recorded on different parts of island shorelines. Average net shoreline movement (NSM) calculated from the transect analysis ranged from 3.71 m per decade on Savave island in Nukufetau to −3.33 m per decade on Fuagea in Funafuti. Collectively, the balance between erosion and accretion on each island yields net changes in island area (Fig. 2b) and also provides the mechanism for effective island migration on the reef platform surfaces as exhibited in planform analysis (Fig. 3). Notably greatest variability in shoreline behaviour occurred on islands located on the rim of larger atolls (Fig. 2), although data confirm that total land area increased in eight of the nine atolls.

## Discussion

Results challenge existing narratives of island loss showing that island expansion has been the most common physical alteration throughout Tuvalu over the past four decades. Of significance, documented increases in island area over this period have occurred as the sea level has been rising. The sea level at the

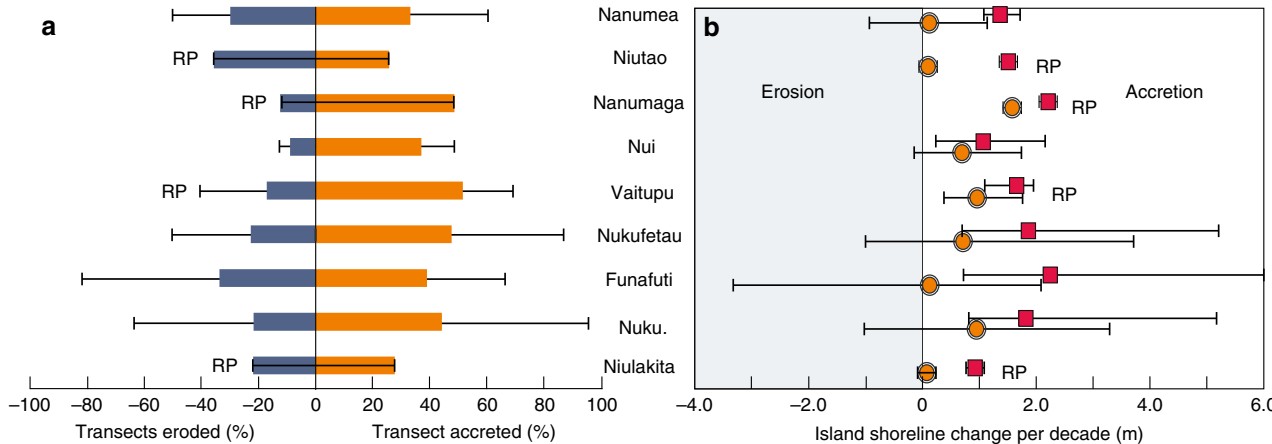

**Fig. 2** Summary changes in shoreline dynamics between atolls based on Digital Shoreline Analysis System analysis of island shoreline transects. **a** Percentage of shoreline transects experiencing erosion (blue bars) and accretion (orange bars) aggregated at the atoll scale, error bars represent maximum transect erosion and accretion in each atoll. **b** Net rate (orange circles) and gross rate (red squares) of shoreline movement per decade aggregated at the atoll scale. Error bars represent minimum and maximum rates within each atoll. Source data: Supplementary Data 2

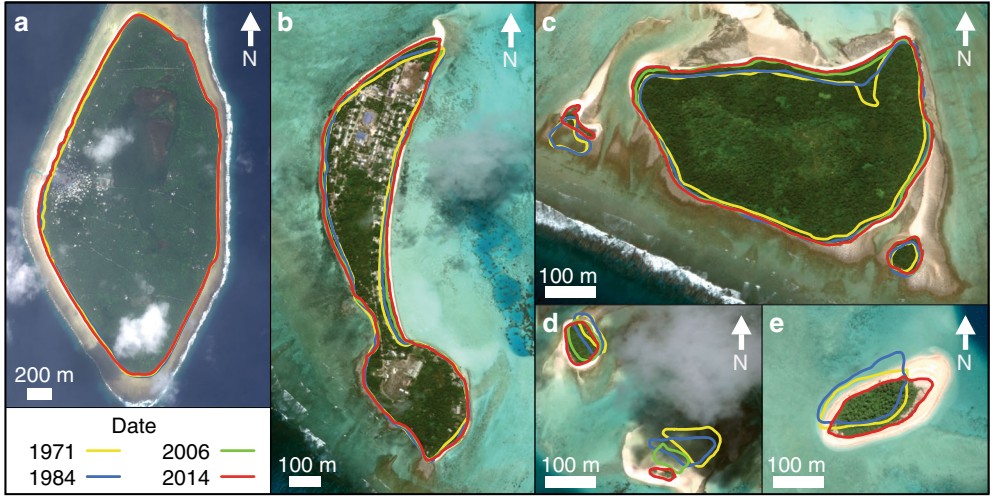

**Fig. 3** Examples of island change and dynamics in Tuvalu from 1971 to 2014. **a** Nanumaga reef platform island (301 ha) increased in area 4.7 ha (1.6%) and remained stable on its reef platform. **b** Fangaia island (22.4 ha), Nukulaelae atoll, increased in area 3.1 ha (13.7%) and remained stable on reef rim. **c** Fenualango island (14.1 ha), Nukulaelae atoll rim, increased in area 2.3 ha (16%). Note smaller island on left Teafuafatu (0.29 ha), which reduced in area 0.15 ha (49%) and had significant lagoonward movement. **d** Two smaller reef islands on Nukulaelae reef rim. Tapuaelani island, (0.19 ha) top left, increased in area 0.21 ha (113%) and migrated lagoonward. Kalilaia island, (0.52 ha) bottom right, reduced in area 0.45 ha (85%) migrating substantially lagoonward. **e** Teafuone island (1.37 ha) Nukufetau atoll, increased in area 0.04 ha (3%). Note lateral migration of island along reef platform. Yellow lines represent the 1971 shoreline, blue lines represent the 1984 shoreline, green lines represent the 2006 shoreline and red lines represent the 2014 shoreline. Images ©2017 DigitalGlobe Inc

Funafuti tide gauge has risen at $3.9 \pm 0.4$ mm y$^{-1}$ over the timeframe of analysis (total rise of ~ + 0.15 m, Supplementary Fig. 3) and this rate of change has been spatially coherent across the archipelago[29]. Results show that there has been no uniform morphological response to this increase in the sea level. While there has been erosion of a subset of smaller-sized islands (~26.5%, Fig. 1), the majority of islands (73.5%) have expanded in area. The absence of a uniform or widespread erosion response indicates that sea-level change alone cannot account for the observed island changes and suggests that there are a set of higher-frequency processes that imprint on island change that may mask the possible effects of incremental sea-level change.

Wave processes and shifts in wave regime have previously been identified as critical controls on island morphological adjustment, and their influence can be expressed in three ways. First, shifts in the incident wave climate may reconfigure depositional nodes on reef surfaces[30]. However, analysis of the 30-year wave hindcast data from the Tuvalu region shows no appreciable change in wave climate since 1979[31,32], implying that this mechanism is unlikely to be responsible for observed island adjustments. Second, rising sea levels can allow a greater transfer of wave energy across reef surfaces, thus enhancing remobilisation of island shorelines and sediment transfer[33–35]. There is compelling evidence to indicate that this process has exerted an influence on atoll rim islands throughout the archipelago, expressed as ocean shoreline erosion and lagoon shoreline accretion (Figs. 3b, c, e) resulting in net lagoonward migration of islands[36,37]. However, it is important to highlight that, in many instances, such migration responses have also been accompanied by island expansion. Third, storm wave processes can influence island morphology and size, although erosion or accretion trajectories vary depending on storm magnitude and the grade of material comprising islands[38,39]. While located outside the primary zone of cyclogenesis, the Tuvalu archipelago is periodically imfluenced by cyclone events that generate wave heights between 3 and $4 + $m[40,41]. In Tuvalu, it is possible that extreme wave events can partly explain spatial differences in observed island change. For example, Cyclone Bebe (1972) delivered significant volumes of coarse sediment to the

Funafuti reef flat, which were subsequently reworked to the island shorelines expanding the footprint of the islands on the eastern rim of Funafuti over the four decades[37,40,42]. Such episodic events and their subsequent constructional effects could account for the predominant expansion mode of mixed sand–gravel and gravel islands elsewhere in the archipelago. In contrast, the same events may have destabilised sand islands. Our data show that 54% (13 of 24) of the sand islands reduced in size over the timeframe of analysis. In Funafuti and Nukufetau these islands are located on the leeward northwest and northern sectors of atoll rims. While construction of the islands has occurred under lower-energy regimes periodic storms may have promoted erosion and destabilisation of these islands.

While wave processes can account for locational shifts in shorelines, they cannot solely account for the expansion of the majority of islands. Expansion of islands on reef surfaces indicates a net addition of sediment. Implications of increased sediment volumes are profound as they suggest positive sediment generation balances for these islands and maintenance of an active linkage between the reef sediment production regime and transfer to islands, which is critical for ongoing physical resilience of islands[43]. Such island reef budgets and their connectivity are likely to be spatially variable as a consequence of the localised reefal provenance of island sediments and the temporal dynamics of reef ecology and sediment generation and transfer mechanisms[37,43,44]. On most windward reef sites such linkages are modulated by storm-driven wave deposition of new materials and subsequent reef recovery, whereas at leeward locations, where sand islands may prevail, supply is likely to be characterised by a more consistent incremental addition of sediments from reef flat surfaces.

Direct anthropogenic transformation of islands through reclamation or associated coastal protection works/development has been shown to be a dominant control on island change in other atoll nations[24,27,45,46]. However, in Tuvalu direct physical interventions that modify coastal processes are small in scale as a consequence of much lower population densities. Only 11 of the study islands have permanent habitation and, of these, only two

islands sustain populations greater than 600. Notably, there have been no large-scale reclamations on Tuvaluan islands within the analysis window of this study (the past four decades). On the most densely urbanised island Fogafale, there has been minimal direct shoreline modification up to 2014[47] with observed increases in island area occurring well beyond the main settlement areas. Elsewhere in the archipelago, direct shoreline modification is also limited in scale and includes coastal protection works along a short length of Savave shoreline in Nukufetau, dredging of boat access channels across reef flats, and construction of associated boat-landing structures. Data suggest that these modifications have had a negligible direct impact on coastal change at the construction sites or adjacent sites alongshore with expansion occurring well outside the footprint of human settlements.

Consequently, documented changes in islands throughout Tuvalu are considered to be driven by environmental rather than anthropogenic processes. In particular, wave and sediment supply processes provide the most compelling explanation for the physical changes documented in islands, most notably the expansion of the majority of islands, and their locational adjustments over the past four decades. Collectively, these processes can mask any incremental effects of rising sea level, making attribution of sea-level effects elusive, as these processes can promote higher frequency and larger magnitude changes in islands that can persist in the geomorphic record.

On the basis of empirical changes in islands we project a markedly different trajectory for Tuvalu's islands over the next century than is commonly envisaged. Observations over the past four decades indicate that the future of Tuvalu's islands will be marked by a continual changing mosaic of physical land resources.

Changes expected include the ongoing erosion of smaller sand islands in the archipelago (<1 ha), continued expansion of the majority of medium (1–10 ha) and larger-sized islands (>10 ha), stability of reef platform islands and increased mobility of atoll reef rim islands. Such changes suggest that the existing footprint of islands on reef surfaces will continue to change, although the physical foundation of islands will persist as potential pedestals for habitation over the coming century. Consequently, while we recognise habitability rests on an additional set of factors[4,11–13] loss of land is unlikely to be a factor in forcing depopulation of islands or the entire nation. However, changes in land resources may still stress population sustainability in the absence of appropriate adaptive initiatives.

Significantly, our results show that islands can persist on reefs under rates of sea-level rise on the order of $3.9 \pm 0.4$ mm yr$^{-1}$ over the past four decades (Supplementary Note 2, Supplementary Fig. 3) equating to an approximate total rise of ~0.15 m. This rate is commensurate with projected rates of sea-level rise across the next century under the RCP2.6 scenario mid-point rate of 4.4 mm yr$^{-1}$ (range 2.8–6.1 mm yr$^{-1}$)[48]. However, under the RCP8.5 the projected rate of sea-level rise will double to 7.4 mm yr$^{-1}$ (range 5.2–9.8 mm yr$^{-1}$). Under these higher sea-level projections it is unclear whether islands will continue to maintain their size, although the dynamic adjustments observed are expected to occur at faster rates placing a premium on establishing ongoing monitoring of island morphological dynamics.

Recognition that land resources will remain through the next century also challenges past and current paradigms of adaptation. It has been argued that the adaptation experience in atoll countries to date has been poor[6,17]. Underpinning past approaches to adaptation have been a set of time to extinction projections, implying that habitability of islands is likely to be severely compromised in the coming decades[4,16]. In part, this is due to the lack of relevant information on the type and scale of changes expected

in the future against which to inform adaptation planning[17]. Without such knowledge adaptation solutions have been captured by the rhetoric of loss, which has foreclosed robust consideration of sustainable adaption options. Our analysis provides an empirical basis to reconceptualise alternate and more creative adaptation pathways in atoll nations with continued habitation of islands underpinning these approaches.

Quantified patterns of island physical dynamics provide a sound basis for new approaches to land use planning. The Tuvalu data indicate that reef platform islands have remained the most stable islands and in most instances have increased in area. However, despite their larger size (>10 ha) and stability these islands remain among the least densely populated. For example, the reef platform islands of Nanumaga (3.0 km$^2$) and Vaitupu (5.18 km$^2$) have population densities of 183 and 297 km$^{-2}$, respectively, which are much lower than the urban island of Fogafale (area of 1.59 km$^2$) with a population density of 3427 km$^{-2}$. Notably, medium-sized islands (1–10 ha) have largely expanded over recent decades and, despite the fact that these islands are scarcely populated, they could provide opportunities for future habitation across the archipelago. Smaller islands appear the most dynamic, in some cases experiencing marked erosion and, therefore, do not provide ideal sites for ongoing habitation.

Insights into island change in Tuvalu parallel observations on biophysical change made elsewhere[25,26,46,49,50] and allow us to reflect more widely on patterns of population distribution and resource pressures in other atoll nations. Current population distributions in atoll nations are legacies of economic and social investment rather than reflective of the carrying capacity of the land and may be considered not well aligned to the changing mosaic of island adjustments observed over the past four decades. Contemporary histories of population movement and settlement in the Pacific are shaped by geopolitical influences on the distribution of economic, transportation, health, educational and livelihood opportunities at a national scale[51]. Commonly, the densest populations are located in the economic and political centres, situated on smaller and less stable islands, which represent less than 1% of the land available in archipelagoes. The complexity of habitability in these settings is also coupled with competing discourses of abandonment, displacement and threats to human security.

Against this backdrop of patterns of human resettlement, exploring opportunities presented by the dynamic mosaic of land availability necessitates a reconsideration of how land-use planning is undertaken that recognises the heterogeneity of island changes[21], existing land tenure systems, patterns of food security[52] and approaches to support internal migration within atoll nations. Such suggestions are by no means novel[14] but to date long-term planning has been constrained by concerns about lack of data about island change to support informed decisions. Here we have presented more compelling evidence that islands may persist and encourage a re-engagement with what alternative adaptation pathways may look like.

If collective narratives are imagining atoll island futures beyond geo-political boundaries, destabilising cultural identity and sovereignty, we ask on the compelling evidence of changing island dynamics and future land availability, is it inappropriate to also re-imagine intranational migration and to consider the development, political and cultural implications of such relocations? To date, such movement in the Pacific has had varied outcomes and been driven by formalised relocation agendas and more informal movement between place, highlighting experiences of cultural and economic disconnection[53,54]. However, it has also been argued that internal relocations can work more effectively and communities experience less trauma where they are familiar with the places they ultimately move to, have time to plan and are

in control of that planning, have time to accommodate the idea of movement and move at a time of their choosing in an orderly manner[55]. Not least at issue here is the requirement for significant and continued economic investment, including the development of opportunities for appropriate economic growth and sustained adaptive capacity.

Embracing such new adaptation pathways will present considerable national-scale challenges to planning, development goals and land tenure systems. However, as the data on island change show there is time (decades) to confront these challenges, which could engender more thoughtful support from international agencies. The pursuit of this and other alternate adaptation pathways does not negate the need to still vigorously support ongoing mitigation action to curtail future sea level impacts and climatic changes on small island nations or to undertake robust efforts to better define the constraints and thresholds of habitability (such as water resources and food supply) on atoll islands. These collective efforts provide a more optimistic set of approaches to adaptation, which support the rights of atoll people to dignified lives and autonomy for future generations and maintaining the sovereignty of atoll nations.

## Methods

**Data sources**. Remotely sensed assessments of shoreline change along coasts within developed nations typically involve the use of temporally rich collections of aerial photographs spanning several decades[56]. However, atoll nations in the Pacific are remote and have limited collections of aerial imagery. Initial imagery from Tuvalu was flown in World War II associated with military occupation of Funafuti atoll. National aerial coverage was first flown in 1971. To examine shoreline change on islands throughout the Tuvalu archipelago, we compare shoreline positions reconstructed from historic aerial photographs captured between 1943 (fragmentary), 1971 and 1984, and high-resolution panchromatic (WorldView-1) and multispectral (QuickBird-2, WorldView-2 and WorldView-3) satellite imagery collected between 2004 and 2015. The principle analysis window (1971–2014) is ~43 years in length.

**Image processing**. Multispectral satellite imagery was pan-sharpened, a process through which the coarser resolution multispectral imagery is sharpened using higher-resolution panchromatic imagery captured simultaneously. The oldest satellite imagery for each atoll provided the source of ground control points for georeferencing imagery. Given the paucity of stable anthropogenic features on most islands, a range of natural features such as cemented conglomerate and beachrock were used as ground control points following similar studies in the Republic of the Marshall Islands[25]. Images were georeferenced in ArcMap and transformed using a second-order polynomial transformation.

**Shoreline interpretation and analysis**. The edge of vegetation is widely used as a proxy for the shoreline within island change studies in atoll settings[24,25,56]. The edge of vegetation is readily identifiable in all imagery, regardless of image colour and contrast and irrespective of environmental conditions such as glare and waves, all of which can impede the interpretation of subtidal and intertidal features such as the toe of beach. The edge of vegetation represents the vegetated core of the island and filters short-term noise associated with the interpretation of dynamic beach shorelines. Where 1971 shorelines are cloud-obscured, preventing the creation of a closed polygon, we use previously calculated areas for the vegetated edge of island[57].

Three sources of uncertainty were considered when calculating the positional uncertainty in edge of vegetation, being rectification, pixel and digitising errors[25]. Rectification error was derived from the Root Mean Square Error of georeferencing. The spatial resolution of scanned aerial photographs and satellite imagery represents the pixel error. The digitising error was calculated as the SD of shoreline position from repeated digitisation of the same section of coast by a single operator. Total shoreline error (Te) was calculated as the root sum of all shoreline positional errors and ranged between 1.31 and 3.46 m.

Shoreline change analysis was undertaken using the Digital Shoreline Analysis System (DSAS) an extension within the GIS software package ArcMap[58]. DSAS analyses change by recording the intersection of transects cast perpendicular to a user-generated baseline and the shorelines. In this study, transects were cast every 10 m along the baseline with a total of 19,403 transects analysed for the archipelago. A range of change statistics were then calculated automatically using the position of the intersection of shorelines and transects. In environments with high temporal resolution records of shoreline positions regression-derived measures of shoreline change rates are widely used[56,59]. However, due to the limited number of shorelines used in this study regression-derived shoreline change rates are unreliable. As a result, two measures of island change are utilised

in this study. First, NSM, the distance between two selected shorelines, was calculated. Second, the annualised rate of change between two shorelines, known as the end point rate (EPR) was calculated. Given the multidecadal timeframe of the data set, the EPR is expressed as decadal rate of change (m per decade). A confidence interval of $2\sigma$ (95.5%) was applied when calculating shoreline change rates. Transects with statistically significant rates of change are considered erosional (–/ve EPR) or accretionary (+/ve EPR), the remaining transects are classified as exhibiting no detectable change.

**Data availability**. All data are contained in Supplementary Information. Source ArcMap shapefiles are available from the authors on request.

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

## Acknowledgements

Research was partially supported by a University of Auckland internal research grant (UOA 3700514).

## Author contributions

P.S.K. conceived the project and led analysis and writing of the manuscript. M.R.F. led remote sensing and DSAS analysis. S.D.O. contributed to the adaptation context for the article and assisted manuscript writing and preparation.

## Additional information

**Competing interests:** The authors declare no competing financial interests.

