## [Peer Review File · Nature Communications]

Editorial Note: This manuscript has been previously reviewed at another journal that is not operating a transparent peer review scheme. This document only contains reviewer comments and rebuttal letters for versions considered at Nature Communications. Mentions of prior referee reports have been redacted.

Reviewers' comments:

Reviewer #1 (Remarks to the Author):

This is a very good and provocative publication that could be improved with a series of minor changes and some attention to three larger issues.

The larger issues are these:

1) the paper plays up the extent to which the problem of habitability is said to be about shoreline changes: yes this is often mentioned in narratives about rising seas, but there are other factors at play that may be just as critical- such as water and food availability. These too may be amenable to treatment by adaptation (in fact easier). At present it will be too easy for critics to claim that the paper misses the point because it plays up only one driver of change (shoreline change) and plays down other drivers. To correct this some slightly more qualified wording in the abstract, and a bit more recognition of other drivers of change and that this paper only focuses on one main driver seems necessary.

2) protected rates of sea level are likely to be far greater than already experienced, so the record thus far may not be very indicative of future change, in which there may be non-linear responses. To direct this, it requires a sentence that compares the sum of past sea-level rise with future project rates, and which acknowledges that the past may not be a perfect guide to future responses. This also suggests qualifying the conclusions, which tend to say 'for the next century', when changes over that timescale seem to uncertain to be so confident about (e.g. line 152)

3) the causes of the observed changes are scarcely considered, yet surely these matter a lot. How much of the accretion is due to construction? what role do storms play? Given most beaches are constructed by foams, what is known about biological productivity now and into the future? and so on... An additional paragraph that draws on any available evidence of causes from Tuvalu or related systems seems necessary and might also help inform thinking about the drivers of change and the scope for adaptation.

Minor corrections (some of which pick up on the above).

Abstract:

line 6, consider changing to 'are key drivers expected to render'...

line 8, consider changing 'physical change' with 'changes in shorelines'

line 9, consider adding a phrase that explains the method (e.g. 'using remote sensing data')

line 10, change to 'local sea level'

line 14, is a bit awkward, could be 'challenge conventional constructions of erosion and inundation and key drivers of migration'.

suggest deleting lines 15, 16, and half of 17, so it reads ...'suggesting scope for new, flexible...'

line 18. consider changing to 'coastal defence'

line 19, reads poorly, consider rewriting.

Para 1. Bit long and obtuse, this could be one sentence.

Lines 38-48. This is long and a bit confusing. The relationship between the science, popular narratives, what is known about adaptation, and what actually happens could be made clearer.

Line 66 - here is where comparison of the sum of past SLR and projected future SLR could be made.

Fig 1: does latitude play a role here? in terms of SST, and storm intensity for example?
line 176, change 'geopolitical discourses' to 'colonial associations'
ref 28, Connell has two 'la's.

Reviewer #2 (Remarks to the Author):

Nature Science – Atoll Nations

The article contributes new, interesting and valuable information on an important issue and should be published. It extends the work done earlier by Webb and Kench (though not enough recognition is given to this, especially where it extends from Kench et al 2015). Certain changes would greatly strengthen the paper. More specifically the morphological analysis is very useful stuff, but their speculations on how outer islands might become destinations for migration within Tuvalu (and by extension within other atoll states) flies in the face of ongoing population movements and seems excessively optimistic. In other words the social science, the newest element here, is weak.

Title – Technically it only provides data for one nation although other studies (inc. Webb and Kench, Ford, Mann and Westphal in Takuu, Duvat in the Tuamotus etc – uncited here) do indicate that changes in Tuvalu are certainly not unique.

It is a generous assertion that it provides 'new opportunities' – 'less pessimism' might be more appropriate....

Line 6 Ref 3 Long before Barnett and Adger – Roy and Connell (Journal of Coastal Research, 1991) pointed this out and this should probably be acknowledged.

Line 13-14 While it is true that the 'conventional constructions' of the media largely continue to point to island loss and the need for adaptation (a) academia has accepted and used Webb and Kench (2010) and a quick check of any citation analysis will reveal that; (b) assertions about the need for adaptation tend to link to cyclone threats, storms and overwash - as much as land loss (despite its symbolic significance). ...See also Line 49.

Line 23 'Historic' is a vague word. What time period? Pre-contact? Pre-war?

L26 What is a 'cultural staging platform'?!

L31. Some think otherwise on threats to livelihoods. Jimmie Rodgers (cited in Connell 2013 Islands at Risk) claims it is NCDs. Others point to basic economic issues, especially in atoll states, like Tuvalu, that have little to export and livelihoods are challenging enough without climate change.

L40. Again, long before Barnett, there were discussions of climate change, security and migration (see Connell, Climatic Change. A New Security Challenge for the Atoll States of the South Pacific, Journal of Commonwealth and Comparative Politics, 1993).

L45-48. This last sentence is quite unclear. To what extent is an archipelagic atoll state – heterogeneous? (see also Line 59). That is surely to be demonstrated, through, presumably, atoll morphology, unless the authors are suggesting other forms? What is meant by the 'historical imprint of colonial agendas'? The authors link this to 'entrenched land tenure systems' – is there a link – and what is the point? I will suggest below that 'entrenched land tenure systems', or some similar phrase, pose one kind of problem for the 'internal migration' that the authors suggest is/may be feasible.

L51. 'A richer set'? Authorial optimism?

L52 'states'

L64. Where does the national population total of 10,600 come from? This seems a considerable over-estimate ...

Oddly - the Wikipedia entry 'Demographics of Tuvalu- is useful background (though it is also the source of the 2012 census data). At least 2000 Tuvalu- born live in NZ and the de facto population of Tuvalu is probably around 9500.

L96. Tepuka spelt Te Puka in supplementary tables.

L98. 'highlights'

L105 'yields'

L106 'provides'

L121 Again – useful to point to the 'existing narratives' of land loss. Cite? Storlazzi? Dickinson? Kelman? Albert et al?

L126 Ref 25. Useful to cite Bayliss-Smith, Physical effects of hurricane Bebe upon Funafati atoll, Tuvalu, Australian Geographer, 1987.

L128 Sentence is rightly cautious about attribution of alterations to particular combinations (?) of ongoing anthropogenic and other influences on morphology but then (L130-131) goes on to predict a 'markedly different trajectory' from that which others have envisaged (Again- who are these straw men?) Where did the caution evaporate?

L151 The claim that islands will remain 'habitable' over the coming century is based on area alone. It takes no account of any possible impact of cyclones and droughts (both of which have unusually influenced Tuvalu in this decade) – let alone any aspirations that Tuvaluans might have for different kinds of lives, or global economic fluctuations.

L166. 'Counter-intuitively .. reef platform islands are the least populated'. That presumes that the atolls are homogeneous (although the authors previously point to heterogeneity). The reef islands either do not have lagoons (or very small enclosed lagoons) that means both that access is often difficult but, more importantly, marine diversity is more limited. That very much limits livelihoods and reduces their potential for resettlement.

L174. 'Ill-matched' – in what sense?

L178 Connell (2003) is strange citation here, even though it refers to socio-economic changes in Tuvalu. Far better to cite Connell, Vulnerable Islands: Climate Change, Tectonic Change, and Changing Livelihoods in the Western Pacific, The Contemporary Pacific (2015), which has a more detailed analysis of population shifts in the atoll states.

The scale of the argument seems to shift here to the atoll states more generally, in which case it is not clear what the evidence is for Tarawa and Majuro being on smaller, less stable islands.

L193-197. It is appropriate to re-imagine (L190) but there are three migration streams in most atoll contexts: (a) resettlement – a failure in Kiribati – either in the Line or Phoenix islands (yet that is the mode that the authors prefer) as it was in the Marshall Islands decades earlier, and re Gilbertese in Solomon Islands, and has only really worked in unusual circumstances in pre-colonial times, and then with cautionary tales- hence, with one minor exception, are no longer being contemplated, (b) urbanisation – with problematic consequences - on which there are multiple studies, (c) international migration – and Tuvalu is following the 'Polynesian model' – with numbers growing steadily in NZ with no growth in Tuvalu. Marshallese migration is even more dramatic. In other words Tuvaluans are shifting from the outer islands (a concept that is missing here) especially and moving into urban centres nationally or internationally. Moving to small resource-poor islands flies in the face of aspirations (even without the land tenure considerations). Connell reviews these resettlement problems in Australian Geographer (2012) and with particular reference to the failures of Carteret Islands resettlement in Asia-Pacific Viewpoint (2016)

By now since my cover has probably been blown, I had the same positive thoughts about the potential for outer island atoll development three decades ago (Pacific Studies, 1986) but the reluctance of people to move to resource-poor islands, without kinship ties and thus access to land, and the inability of government to develop – or even maintain – infrastructure for these islands has proved me wrong. What Wilmsen (sp) and Webber say about resettlement (even based on China) is entirely valid, but the converse is that the 'recipients' of this planning have little interest in developing plans. The first sentence of the next paragraph (L198-199) is thus entirely true ..but it is not just at national scale.

L292. Citation refers to RMI hence sentence needs some clarification.

Reviewer #3 (Remarks to the Author):

REVIEW NATURE COMMUNICATIONS 2017.

Changing Mosaics of Habitability Provide New Opportunities for Adaption in Atoll Nations

This is a clearly written paper stating a case study on Tuvalu where decadal shoreline analyses yield surprising results about island change. The authors did an incredibly thorough analysis and found that most islands had increased their area. The consequences that this finding has for the present and future management of atoll nations are then discussed, suggesting that "existing narratives of island loss" are not applicable and that more thorough analyses of island change should be undertaken to identify the most stable islands where to establish the nations' populations.

The claims are novel enough and they bring a message that should be emphasised and published. This paper will encourage discussion on how to manage the effects of climate change in atoll nations; something that is needed, and it comes in a timely manner.

The paper lacks discussion on some issues and it would need further clarification on some others.

Below is a list of themes that I think should be mentioned in the paper; while I understand that they might be well beyond the scope of this paper, I think that they should at least be mentioned so the readers understand the uncertainty that the future might bring:

1. Definition of island. What are the minimum dimensions of a shoal to be considered an island? What are the characteristics? Given the methods, I assume that the authors only measured vegetated islands but that should be clarified in the manuscript.
2. The authors discuss different types of islands and how ones are more stable than others; island types should be clearly defined. Perhaps another colour could be used for island types in Supplementary Figure 2. Atolls should be somehow distinguished from platforms in the tables and figures in the main manuscript. That would help the readers follow the discussion in the manuscript more easily.
3. The opening statement in Page 7 imply that the authors expect a continuity in the changes that they have measured; I assume that they are taking into account a constant rate of sea level rise. What would happen if rates of sea level rise accelerate? There are examples in the literature showing that some islands within The Solomon Islands underwent extreme erosion under faster sea level rise.
4. The sediment sources are not discussed. I wonder whether bleaching due to warming would mean that there will be less sediment in the future. I do acknowledge that this is not the scope of this paper but a couple of sentences acknowledging the uncertainty would soften some of the claims and open the discussion towards future steps in research of island stability.
5. Still on the sediments, is there a relation between the island sediment composition and its stability or tendency to erode or accrete?
6. What about other effects of climate change like, for example, changes in storm frequency, direction, and intensity? The studied period has seen several cyclones in the area (e.g., Bebe) that caused dramatic changes in the area.
7. What about the wave/wind climate in the area? Is the study period representative of a wide range of conditions including climatic cycles that might affect wave and wind direction? I assume this is the case, given the length of the study period, however the authors should state it. Further analyses could be made considering the degree of exposure to wave energy of the erosional vs accretional transects.

8. It is not clear whether the erosion that has happened has affected essential infrastructure for the islands' inhabitants. I feel that this should be stated somewhere in the paper as the islanders might not care whether the island is growing in one side if they are losing their infrastructure on the other.

9. Why is that platform islands are more resilient? And why is that they are less populated? Is there a socio-economic reason (e.g., less resources, water...)?

10. Supplementary figure 2 helps a great deal when understanding the discussion of the paper. I think that aside from stating the types of islands, the figure should show which are the inhabited islands and which islands hold the most important cities/infrastructure. Equally, this figure should show the spatial distribution of the accreting, stable, and eroding islands by using different colours or markers. This addition will help understanding the discussion about the socio-economic and cultural costs of relocating the nation's inhabitants. Marking all inhabited islands might be difficult or next to impossible to do for all the studied islands given the scale of the figure but pointing out the main cities should be achievable.

11. Finally, how representative is Tuvalu of the possible evolution of other atoll nations?

The manuscript is clearly written and it is easy to understand. The methodology is straightforward and standard and could be easily replicated. All the methods seem sound. Here are a few other comments/details about the paper:

1. I don't think the title is clear enough, the term mosaics of habitability sounds more like a GIS tool and does not send a clear message.

2. Abstract:

2a. Please add "locally" to the statement about the sea level rising three times the global average.

2b. The results' highlights are three points, the first two seem to be the same point. Please revise punctuation in third point, it should be "; and,

3. The text is clear but there is a bit of wordiness to it, for example, at the end of Page 1 the sentence "islands have been conceptualised as cultural staging platforms" is not very clear and could be worded otherwise." The same goes to the last sentence on page 6 "continual changing mosaic of physical land resources"

4. Use of word spectre on the second page; I am guessing the authors mean spectra.

5. Figure 3 needs arrows indicating the lagoonward direction, it is not clear to the reader.

6. Supplementary Figure 1 needs a world map or something to put everything in context.

7. There is a typo on the * in Supplementary Table 2, it says gavel instead of gravel. How was the island sediment composition obtained? Visually? Please state it.

Ana Vila-Concejo
Sydney, 04.08.2017

Reviewer #1 (Remarks to the Author):

We thank the reviewer for their positive and constructive comments on the manuscript noting they see the study as provocative and good publication. The reviewer raises three issues that we address below and we outline how we have modified the manuscript to account for the comments. The review comments are in bold text followed by our response.

1) The paper plays up the extent to which the problem of habitability is said to be about shoreline changes: yes this is often mentioned in narratives about rising seas, but there are other factors at play that may be just as critical- such as water and food availability. These too may be amenable to treatment by adaptation (in fact easier). At present it will be too easy for critics to claim that the paper misses the point because it plays up only one driver of change (shoreline change) and plays down other drivers. To correct this some slightly more qualified wording in the abstract, and a bit more recognition of other drivers of change and that this paper only focuses on one main driver seems necessary.

In large part we agree with this comment. We would point out that the majority of articles on this topic also target a single driver of habitability (i.e. inundation or loss of potable water). Also of note is the still persistent assumption that land will inevitably be lost. There are still few datasets that examine the land dynamics aspect of the problem and future persistence is still regularly called into question. The novelty of this manuscript in part rests on the fact we have examined every island in the nation rather than a small subset.

However, we fully accept the argument raised by the reviewer and have included a statement early in the manuscript (lines 44-49) indicating that habitability is dependent on a number of factors (food availability, water, susceptibility to hazards, socio-political factors). We have added references that also capture this point (Connell, 2015; McCubbin et al., 2015; Marotzke et al., 2017). We then argue that this manuscript provides a robust analysis of one of the underpinning factors – land availability.

2) Protected rates of sea level are likely to be far greater than already experienced, so the record thus far may not be very indicative of future change, in which there may be non-linear responses. To direct this, it requires a sentence that compares the sum of past sea-level rise with future project rates, and which acknowledges that the past may not be a perfect guide to future responses. This also suggests qualifying the conclusions, which tend to say 'for the next century', when changes over that timescale seem too uncertain to be so confident about (e.g. line 152).

We agree with this comment and have made a number of adjustments to the manuscript and supplementary information to appropriately deal with this issue:

- i) We have included the sea level data from Funafuti atoll, the only site with a sea level gauge within the archipelago, to provide a framework for discussion on this point (see new supplementary Figure 3). We make this point and refer to the new figure in lines 80-82 and 146-148).
- ii) The sea level data shows there has been a 3.9 ± 0.4 mm/yr increase in sea level over the timeframe of analysis (1977-present) which corresponds to the period of analysis of island change (40 years). Of note, this rate of rise equates to

approximately 0.4m/100 yrs, which is comparable to the lower-mid range of projections for the next century. We have inserted a sentence outlining this recent rapid rate of sea level change in the archipelago at the end of the discussion.

iii) We now discuss our findings in light of future changes in sea level in lines 247-255. We note that:

'Significantly, our results show that islands can persist on reefs under rates of sea level rise on the order of 3.9 ± 0.4 mm/yr over the past four decades (Supplementary Figure 3) equating to an approximate total rise of 0.15 m. This rate is commensurate with projected rates of sea level rise across the next century under the RCP2.6 scenario mid-point rate of 4.4 mm/yr (range 2.8-6.1 mm/yr)⁴⁸. However, under the RCP8.5 the projected rate of sea level rise will double to 7.4 mm/yr (range 5.2-9.8 mm/yr). Under these higher sea level projections it is unclear whether islands will continue to maintain their size, though the dynamic adjustments observed are expected to occur at faster rates placing a premium on establishing ongoing monitoring of island morphological dynamics.'

3) *The causes of the observed changes are scarcely considered, yet surely these matter a lot. How much of the accretion is due to construction? what role do storms play? Given most beaches are constructed by forams, what is known about biological productivity now and into the future? and so on... An additional paragraph that draws on any available evidence of causes for Tuvalu or related systems seems necessary and might also help inform thinking about the drivers of change and the scope for adaptation.*

We agree and have now inserted a substantive section in the discussion (new lines 155-217) to examine key environmental processes that promote change and the likely influence of these process in changing islands within the archipelago. The section now considers, sea level, wave regime, storms and sediment supply in accounting for island change. We do note that most beaches are not composed of foraminifera as suggested by the reviewer but rather coral and coralline algae.

Minor corrections (some of which pick up on the above).

Abstract:

- ***line 6, consider changing to 'are key drivers expected to render'...***

Changed. Though the abstract has now been rewritten to conform to the Nature Communications format.

- ***line 8, consider changing 'physical change' with 'changes in shorelines'***

Changed to 'shoreline change'.

- ***line 9, consider adding a phrase that explains the method (e.g. 'using remote sensing data')***

Change instituted 'Using remotely sensed data...'. Abstract now modified to conform to the Nature Communications format.

- ***line 10, change to 'local sea level'***

Instituted change.

- **line 14, is a bit awkward, could be 'challenge conventional constructions of erosion and inundation and key drivers of migration'.**

Yes – but we feel the results have much deeper implications for approaches to adaptation that include possibilities other than migration. We have inserted 'erosion and loss'. We are unable to extrapolate assertions of changes to inundation from the dataset we present.

- **suggest deleting lines 15, 16, and half of 17, so it reads ...'suggesting scope for new, flexible...'**

We have reworked the abstract to comply with the Nature Communications format. The changes have taken into account the suggested deletion.

- **line 18. consider changing to 'coastal defence'**

We have instituted this suggestion.

- **line 19, reads poorly, consider rewriting.**

We have reworked the abstract to comply with the Nature Communications format and therefore, have reworked this sentence

- **Para 1. Bit long and obtuse, this could be one sentence.**

We have rewritten the first paragraph in order to clarify and simplify.

- **Lines 38-48. This is long and a bit confusing. The relationship between the science, popular narratives, what is known about adaptation, and what actually happens could be made clearer.**

We have rewritten this section in order to clarify the points made.

- **Line 66 - here is where comparison of the sum of past SLR and projected future SLR could be made.**

We have responded to the issue related to sea level and have inserted discussion on past vs future sea level change (lines 247-255).

- **Fig 1: does latitude play a role here? in terms of SST, and storm intensity for example?**

We have now included a substantive discussion component concerning the environmental drivers of change (new lines 155-217). This section deals directly with storm processes, a function of latitude.

- **line 176, change 'geopolitical discourses' to 'colonial associations'**

We understand that there are a broader set of influences at work in shaping population movement and settlement. A colonial imprint is certainly important in influencing the development of infrastructure and nodes of settlement but there are a wider set of geopolitical drivers at play. We have amended this sentence to refer to geopolitical influences rather than discourses.

- **ref 28, Connell has two 'la's.**

Typographical error has been corrected.

Reviewer #2 (Remarks to the Author):

We thank the reviewer for their insightful comments on our manuscript. We note they indicate the article contributes new, interesting and valuable information on an important issue and should be published. The reviewer makes a number of useful suggestions to improve the manuscript and we respond to these suggestions in detail below. The review comments are in bold text followed by our response.

Title – Technically it only provides data for one nation although other studies (inc. Webb and Kench, Ford, Mann and Westphal in Takuu, Duvat in the Tuamotus etc – uncited here) do indicate that changes in Tuvalu are certainly not unique.

It is a generous assertion that it provides ‘new opportunities’ – ‘less pessimism’ might be more appropriate....

We have altered the title to be more specific about the data contribution and temper our ‘optimistic’ view point as suggested by the reviewer. We also note that we have restructured the discussion to separate the results from our analysis of all islands in Tuvalu from a discussion of the implications for atoll nations in general. This has allowed us to include additional island studies within the manuscript.

Line 6 Ref 3 Long before Barnett and Adger – Roy and Connell (Journal of Coastal Research, 1991) pointed this out and this should probably be acknowledged.

We agree. We are aware of this work but were constrained in the initial formatting re: number of references allowed. We have now included this work in the manuscript within the referencing allocation permitted.

Line 13-14 While it is true that the ‘conventional constructions’ of the media largely continue to point to island loss and the need for adaptation (a) academia has accepted and used Webb and Kench (2010) and a quick check of any citation analysis will reveal that; (b) assertions about the need for adaptation tend to link to cyclone threats, storms and overwash - as much as land loss (despite its symbolic significance). ...See also Line 49.

While we partly agree with this comment there is still evidence in the scientific literature that belief in island loss prevails (Dickinson, Hubbard et al., 2015; Albert et al., 2016; Barnett et al., 2017). For example, at a leading sea level conference in New York in June 2017 sea level rise was considered synonymous with island disappearance. While there have been a number of articles produced in recent years identifying that physical island loss may not occur (beginning with Webb and Kench), acceptance has largely been acknowledged by a small group of island geoscientists and island social scientists. Beyond this group acceptance of the results remains limited. As one of the primary authors of the Webb and Kench article (and subsequent articles) I have received a steady stream of criticism that the findings are wrong – despite lack of any evidence to refute our work. Secondly, our work has been critiqued as we have only examined a small subset of islands, which limits evaluation of adaptation implications to a few locations. Our intent with this article is to present analysis of an entire nation and highlight that consideration of island changes across the entire network of islands (land resources) provides a powerful platform for reconsideration of adaptation options. Our intent was also that publication in a high impact journal would begin to normalise the acceptance of island change/persistence.

Line 23 'Historic' is a vague word. What time period? Pre-contact? Pre-war?

We have clarified the timeframe in the text.

L26 What is a 'cultural staging platform'?!

We have amended this term to more specifically refer to pedestals for human occupation to reflect the anthropological focus on human patterns of settlement and resource use.

L31. Some think otherwise on threats to livelihoods. Jimmie Rodgers (cited in Connell 2013 Islands at Risk) claims it is NCDs. Others point to basic economic issues, especially in atoll states, like Tuvalu, that have little to export and livelihoods are challenging enough without climate change.

We have amended the sentence to indicate climate change is one of the greatest environmental threats to small islands. We also note that the reference used here was the Pacific Forum leaders statement.

L40. Again, long before Barnett, there were discussions of climate change, security and migration (see Connell, Climatic Change. A New Security Challenge for the Atoll States of the South Pacific, Journal of Commonwealth and Comparative Politics, 1993).

We agree and have added references Connell, J 1993. Climatic change: A new security challenge for the atoll states of the South Pacific, *The Journal of Commonwealth and Comparative Politics*. Vol 31 no 2, pp 173-192.

L45-48. This last sentence is quite unclear. To what extent is an archipelagic atoll state – heterogeneous? (see also Line 59). That is surely to be demonstrated, through, presumably, atoll morphology, unless the authors are suggesting other forms? What is meant by the 'historical imprint of colonial agendas'? The authors link this to 'entrenched land tenure systems' – is there a link – and what is the point? I will suggest below that 'entrenched land tenure systems', or some similar phrase, pose one kind of problem for the 'internal migration' that the authors suggest is/may be feasible.

We have reworked the lines to improve clarity. We agree that land tenure is and will be a significant difficulty in nations. Nevertheless, we believe it important to highlight that land tenure perhaps does need to be confronted in a setting where land resources are changing and will continue to constrain adaptation solutions. We have reflected on this point later in the manuscript (new lines 62, 297, 318).

L51. 'A richer set'? Authorial optimism?

We have reworded this phrase as:

'We argue that indeed there are a more nuanced set of options to be explored to support adaptation in atoll states.'

L52 'states' – we have corrected the spelling

L64. Where does the national population total of 10,600 come from? This seems a considerable over-estimate ... Oddly - the Wikipedia entry 'Demographics of Tuvalu- is useful background (though it is also the source of the 2012 census data). At least 2000 Tuvalu- born live in NZ and the de facto population of Tuvalu is probably around 9500.

We obtained the population data from the official statistics which were analysed by Dr Ward Freisen, (Tuvalu Central Statistics Division, 2012 Census of Population and Housing) and downloadable from the following link <http://tuvalu.poggis.spc.int/>.

We note that the 2002 census data had a resident population of approximately 9,500.

L96. Tepuka spelt Te Puka in supplementary tables.

Corrected spelling to ensure consistency.

L98. 'highlights' Corrected spelling.

L105 'yields' Corrected spelling.

L106 'provides' Corrected spelling.

L121 Again – useful to point to the 'existing narratives' of land loss. Cite? Storlazzi? Dickinson? Kelman? Albert et al?

We have significantly expanded this section in response to the other two reviewer's comments (line 155-217). This has afforded the opportunity to discuss the mechanisms responsible for the detected island change. As a consequence we have included a number of additional references, including those identified by the reviewer.

L126 Ref 25. Useful to cite Bayliss-Smith, Physical effects of hurricane Bebe upon Funafati atoll, Tuvalu, Australian Geographer, 1987.

Bayliss-Smith's 1988 article was largely based around observations in the Solomon Islands. In the revisions we have included articles describing the impact of hurricane Bebe on the atoll (Maragos et al., 1973; and Baines and McLean, 1976). We have also included the article by Fitchett – which was published in Australian Geographer.

L128 Sentence is rightly cautious about attribution of alterations to particular combinations (?) of ongoing anthropogenic and other influences on morphology but then (L130-131) goes on to predict a 'markedly different trajectory' from that which others have envisaged (Again- who are these straw men?] Where did the caution evaporate?

L128 was summarising some of the difficulties in assigning environmental drivers to the observed patterns of change in island shorelines. Of note, we have now considerably expanded this section to discuss the different drivers of change and their possible influence (lines 155-217).

Line 130-131 (now 220-223) is based on the fact that our aggregate results of island expansion differ from the still commonly asserted notion that islands will disappear. However, we further note that wider factors shaping habitability need to be considered (lines 243-246).

L151 The claim that islands will remain 'habitable' over the coming century is based on area alone. It takes no account of any possible impact of cyclones and droughts (both of which have unusually influenced Tuvalu in this decade) – let alone any aspirations that Tuvaluans might have for different kinds of lives, or global economic fluctuations.

We agree with this comment in part and have amended the sentence to be more precise that the changes will provide the physical foundation for habitation. We argue that the physical presence of land is an essential precursor for habitation. Cyclones and droughts can and do impact communities. Whether they render islands uninhabitable is questionable. In Tuvalu communities have experienced and withstood multiple cyclone and drought events in the past, which provides an analogue to suggest islands do remain habitable in the medium to long-term. Other external factors such as global economic fluctuations are certainly influences on whether populations may remain or move. However, that is factor well beyond the focus of this manuscript.

L166. 'Counter-intuitively .. reef platform islands are the least populated'. That presumes that the atolls are homogeneous (although the authors previously point to heterogeneity). The reef islands either do not have lagoons (or very small enclosed lagoons) that means both that access is often difficult but, more importantly, marine diversity is more limited. That very much limits livelihoods and reduces their potential for resettlement.

We have amended the text to clarify this point. While we understand the resource constraints that shape livelihood decisions we would suggest, alongside the reviewer's other observation that contemporary (and aspirational) lifestyles of Pacific communities can be different from historic practices. For example, Vaitupu is an island that has experienced past infrastructure investment to support educational and economic opportunities.

L174. 'Ill-matched' – in what sense?

We feel the explanation in the paragraph is consistent with the topic sentence. We identify that the greatest population concentrations are on smaller and more dynamic islands. We have amended this sentence to further clarify this point.

L178 Connell (2003) is strange citation here, even though it refers to socio-economic changes in Tuvalu. Far better to cite Connell, Vulnerable Islands: Climate Change, Tectonic Change, and Changing Livelihoods in the Western Pacific, The Contemporary Pacific (2015), which has a more detailed analysis of population shifts in the atoll states. The scale of the argument seems to shift here to the atoll states more generally, in which case it is not clear what the evidence is for Tarawa and Majuro being on smaller, less stable islands.

We have amended the reference according to the reviewer's suggestion.

L193-197. It is appropriate to re-imagine (L190) but there are three migration streams in most atoll contexts: (a) resettlement – a failure in Kiribati – either in the Line or Phoenix islands (yet that is the mode that the authors prefer) as it was in the Marshall Islands decades earlier, and re Gilbertese in Solomon Islands, and has only really worked in unusual circumstances in pre-colonial times, and then with cautionary tales-hence, with one minor exception, are no longer being contemplated, (b) urbanisation – with problematic consequences - on which there are multiple studies, (c) international migration – and Tuvalu is following the ‘Polynesian model’ – with numbers growing steadily in NZ with no growth in Tuvalu. Marshallese migration is even more dramatic. In other words Tuvaluans are shifting from the outer islands (a concept that is missing here) especially and moving into urban centres nationally or internationally. Moving to small resource-poor islands flies in the face of aspirations (even without the land tenure considerations). Connell reviews these resettlement problems in Australian Geographer (2012) and with particular reference to the failures of Carteret Islands resettlement in Asia-Pacific Viewpoint (2016)

By now since my cover has probably been blown, I had the same positive thoughts about the potential for outer island atoll development three decades ago (Pacific Studies, 1986) but the reluctance of people to move to resource-poor islands, without kinship ties and thus access to land, and the inability of government to develop – or even maintain – infrastructure for these islands has proved me wrong. What Wilmsen (sp) and Webber say about resettlement (even based on China) is entirely valid, but the converse is that the ‘recipients’ of this planning have little interest in developing plans. The first sentence of the next paragraph (L198-199) is thus entirely true ..but it is not just at national scale.

We largely agree with the above comments. Is the time right with heightened concern to re-examine this possibility? The driver of migration is frequently economic and related to livelihood and family aspirations. The resettlement to resource poor islands is certainly problematic (even just from an economic perspective) if these relocations are not reflective of community wants, underpinned by viable infrastructure that is well supported and the tyranny of geography (the ability to move efficiently and cost effectively between islands) is redressed. These commentaries are outside the current scope of our work but we would encourage a re-engagement with what alternative pathways may look like. While acknowledging the cultural complexities, not least at issue here is the requirement for significant and sustained economic investment, including the opportunities for economic growth and sustained adaptive capacity. Small scale community based adaptation projects may fail to support significant or transformative improvements in livelihoods, however the collective investment in projects in the Pacific is not insignificant. Coupled with wider philanthropic and industry interest there may be opportunities to pursue more forward looking dialogues about what at times are depicted as seemingly insurmountable barriers.

L292. Citation refers to RMI hence sentence needs some clarification.

Amended sentence to indicate following the methods adopted in the study from RMI.

Reviewer #3 (Remarks to the Author):

We thank the reviewer for their positive and constructive comments. Of note they highlight the thorough analysis and surprising results and are supportive of the manuscript being published. The reviewer suggests additional discussion and some points of clarification are warranted and we outline our response to these points below. The review comments are in bold text followed by our response.

1. Definition of island. What are the minimum dimensions of a shoal to be considered an island? What are the characteristics? Given the methods, I assume that the authors only measured vegetated islands but that should be clarified in the manuscript.

The analysis adopts the edge of vegetation to define the stable section of islands. This is clearly articulated in the methods (lines 352-357). This definition has become widely adopted in island change studies as it reflects the more stable core of an island that might change in position at timescales longer than seasonal fluctuations in beach position. In addition it is such vegetated islands that communities select to inhabit, rather than unvegetated and highly changeable sand shoals.

2. The authors discuss different types of islands and how ones are more stable than others; island types should be clearly defined. Perhaps another colour could be used for island types in Supplementary Figure 2. Atolls should be somehow distinguished from platforms in the tables and figures in the main manuscript. That would help the readers follow the discussion in the manuscript more easily.

We agree and have made the following amendments to the manuscript.

- i) We have colour coded islands that have accreted and those that have eroded in the Supplementary Figure 2.
- ii) Reef platform islands are denoted by square symbols in Figure 1 and atoll rim islands are denoted by solid circles. In addition we have placed a light blue circle around islands that have significant habitation. We have added text to the caption to highlight these differences.
- iii) We have differentiated reef platform islands in Figure 2 by placing RP next to relevant data lines.

3. The opening statement in Page 7 imply that the authors expect a continuity in the changes that they have measured; I assume that they are taking into account a constant rate of sea level rise. What would happen if rates of sea level rise accelerate? There are examples in the literature showing that some islands within The Solomon Islands underwent extreme erosion under faster sea level rise.

We have amended a discussion to reflect expected changes as a consequence of increased sea level change. This section occurs in the new lines 247-255. We now explicitly discuss the current rates of change in the context of recent sea level and consider the implications for accelerated rates of sea level change into the future against the IPCC global projections.

4. *The sediment sources are not discussed. I wonder whether bleaching due to warming would mean that there will be less sediment in the future. I do acknowledge that this is not the scope of this paper but a couple of sentences acknowledging the uncertainty would soften some of the claims and open the discussion towards future steps in research of island stability.*

As noted by the reviewer a full examination of sediment supply is well beyond the scope of this article and few studies have attempted to examine this important issue. However, we have now inserted a discussion of the potential drivers of island change. In this discussion we consider the role of sediment delivery to islands, particularly that driven by storms, and its importance in influence observed changes in islands (new lines 155-194).

5. *Still on the sediments, is there a relation between the island sediment composition and its stability or tendency to erode or accrete?*

There is no clear relationship between island sediment composition and stability. We do see that some islands in the NW of two atolls have eroded and a number of these are composed primarily of sand. However, this is not consistent and we have a number of gravel islands that have also reduced in size.

6. *What about other effects of climate change like, for example, changes in storm frequency, direction, and intensity? The studied period has seen several cyclones in the area (e.g., Bebe) that caused dramatic changes in the area.*

We agree. We have now included a discussion of the effect of multiple environmental processes in driving observed changes in islands. This includes, wave processes, sea level, storms and sediment supply (new lines 155-217). We make specific mention of the cyclone Bebe and its subsequent impact on island change that occurred over the ensuing decades. We note that projections of changes in storm frequency and intensity are very unclear for this sector of the Pacific making such discussion very speculative as the impacts on islands are not direct and involve spatial differences depending on reef condition, which controls sediment supply, and consequently might yield contrasting geomorphic outcomes. In summary, we have included a more full discussion of environmental processes and explored the nuances of how they can influence island change based on our observations.

7. *What about the wave/wind climate in the area? Is the study period representative of a wide range of conditions including climatic cycles that might affect wave and wind direction? I assume this is the case, given the length of the study period, however the authors should state it. Further analyses could be made considering the degree of exposure to wave energy of the erosional vs accretional transects.*

The revised article now directly addresses the wave regime in the archipelago and potential shifts in wave direction (lines 155-181). Of note, studies have shown there has been no appreciable adjustment in either the direction or magnitude of waves since 1979 in this region (Trenham et al., 2014; Bosserelle et al., 2015).

8. *It is not clear whether the erosion that has happened has affected essential infrastructure for the islands' inhabitants. I feel that this should be stated somewhere in the paper as the islanders might not care whether the island is growing in one side if they are losing their infrastructure on the other.*

We are able to report that patterns of change have had no direct impact on 'essential infrastructure'. Indeed, construction of essential infrastructure arguably may have influenced island change. We have added a section to the discussion in which we explore anthropogenic actions in the context of our island change data (new lines 195-209). We have also added a sentence indicating patterns of change have not directly impacted essential infrastructure to this point.

9. *Why is that platform islands are more resilient? And why is that they are less populated? Is there a socio-economic reason (e.g., less resources, water...)?*

This is a good question and one that is stimulating our ongoing research programme. Our working assumption is that wave processes (refraction) trap sediments on reef platform surfaces rather than potentially driving them across reef platforms to fill lagoons (as is the case in atolls, or reef with substantive lagoons). We have now alluded to this in the discussion, but feel we lack definitive evidence to substantiate this claim at this time.

With respect to population density we do explore the political and socio-economic factors for such differences. However, arguably the water resources on such islands are better than atoll rim islands, though lagoonal resources are absent on reef platform islands. The spatial pattern of population distribution likely reflects historical decisions (including those factors identified above). However, contemporary communities live in a different context (some are less reliant on natural resources) and this can alter considerations of habitation.

10. *Supplementary figure 2 helps a great deal when understanding the discussion of the paper. I think that aside from stating the types of islands, the figure should show which are the inhabited islands and which islands hold the most important cities/infrastructure. Equally, this figure should show the spatial distribution of the accreting, stable, and eroding islands by using different colours or markers. This addition will help understanding the discussion about the socio-economic and cultural costs of relocating the nation's inhabitants. Marking all inhabited islands might be difficult or next to impossible to do for all the studied islands given the scale of the figure but pointing out the main cities should be achievable.*

We agree with the reviewer's suggestion and have made the following amendments to Supplementary Figure 2.

- i) We have colour coded islands that have accreted from those that have eroded.
- ii) We have identified inhabited islands and their populations by inserting circular symbols with population values within them, which are pinned to the relevant island. We believe that the population values are diagnostic of major population centres.

11. *Finally, how representative is Tuvalu of the possible evolution of other atoll nations?*

It is a little unclear what this comment refers to. In terms of evolution the Tuvaluan archipelago is representative of developmental modes observed in other atoll settings. Furthermore, our results are consistent with studies of other atoll islands, though these

have only presented a very small sub-sample of islands within an archipelago. Consequently, on the basis of some supportive data, we believe the Tuvalu data is representative of changes taking place in other atoll nations. We have amended the latter sections of the discussion to highlight the generalities that can be extended to other atoll archipelagos.

Detailed Comments:

1. I don't think the title is clear enough, the term mosaics of habitability sounds more like a GIS tool and does not send a clear message.

We have modified the title to improve clarity.

2. Abstract:

2a. Please add "locally" to the statement about the sea level rising three times the global average.

We have changed the abstract according to the reviewer's suggestion.

2b. The results' highlights are three points, the first two seem to be the same point. Please revise punctuation in third point, it should be ";and,

We have taken the reviewer's suggestion into account in revising the entire abstract.

3. The text is clear but there is a bit of wordiness to it, for example, at the end of Page 1 the sentence "islands have been conceptualised as cultural staging platforms" is not very clear and could be worded otherwise." The same goes to the last sentence on page 6 "continual changing mosaic of physical land resources"

We have taken the reviewer's suggestions into account in reworking the sections identified by the reviewer. We believe/hope the sections are now more clear.

4. Use of word spectre on the second page; I am guessing the authors mean spectra.

No, we intended to use the word spectre, which is appropriate in the context it is written.

5. Figure 3 needs arrows indicating the lagoonward direction, it is not clear to the reader.

We agree. Arrows have been inserted.

6. Supplementary Figure 1 needs a world map or something to put everything in context.

We agree. We have included a regional scale map as panel A in Supplementary Figure 1.

7. There is a typo on the * in Supplementary Table 2, it says gavel instead of gravel. How was the island sediment composition obtained? Visually? Please state it.

We have corrected the typographical mistake.

Island sediment composition was obtained from detailed descriptions and analysis of sediments from each island published by McLean and Hosking (1992). We have included this reference at the foot of the Table.

Reviewers' Comments:

Reviewer #1:

Remarks to the Author:

The manuscript is significantly improved and now suitable for publication.

Reviewer #2:

Remarks to the Author:

I am entirely happy that the the authors have taken note of my own comments, and, as far as I can tell, have responded more than adequately to the suggestions of the other referees. It is an excellent article and should now be published ASAP.

Reviewer #3:

Remarks to the Author:

I believe the authors have done an excellent job in addressing my concerns and the suggestions from the other reviewers. The paper is now stronger and clearer and I only have a few details that can be rapidly addressed (or not) as they are not major.

1. I still don't like the word mosaics in the title but it is not my paper and if the authors and editors think that is good, I will have to agree to disagree.
2. Some figures are in Ha but then the manuscript is in Km², I think that the authors should use the same units along the manuscript as it is confusing. I did not notice this in the first draft!
3. Line 21 insert "at" twice the global average
4. Line 26-27, in context with the discussion you should say which IPCC scenario you are referring to when speaking of the next century forecasts
5. L239 and elsewhere - quantify what you understand as large and medium islands.
6. L312 - delete to after have... move to, have to time...
7. L320, I would remove donor and just leave international agencies as this probably is a question of international justice/responsibility to help out island nations

Reviewer #1 (Remarks to the Author):

We thank the reviewer for their positive and constructive comments on the manuscript noting they see the study as provocative and good publication. The reviewer raises three issues that we address below and we outline how we have modified the manuscript to account for the comments. The review comments are in bold text followed by our response.

1) The paper plays up the extent to which the problem of habitability is said to be about shoreline changes: yes this is often mentioned in narratives about rising seas, but there are other factors at play that may be just as critical- such as water and food availability. These too may be amenable to treatment by adaptation (in fact easier). At present it will be too easy for critics to claim that the paper misses the point because it plays up only one driver of change (shoreline change) and plays down other drivers. To correct this some slightly more qualified wording in the abstract, and a bit more recognition of other drivers of change and that this paper only focuses on one main driver seems necessary.

In large part we agree with this comment. We would point out that the majority of articles on this topic also target a single driver of habitability (i.e. inundation or loss of potable water). Also of note is the still persistent assumption that land will inevitably be lost. There are still few datasets that examine the land dynamics aspect of the problem and future persistence is still regularly called into question. The novelty of this manuscript in part rests on the fact we have examined every island in the nation rather than a small subset.

However, we fully accept the argument raised by the reviewer and have included a statement early in the manuscript (lines 44-49) indicating that habitability is dependent on a number of factors (food availability, water, susceptibility to hazards, socio-political factors). We have added references that also capture this point (Connell, 2015; McCubbin et al., 2015; Marotzke et al., 2017). We then argue that this manuscript provides a robust analysis of one of the underpinning factors – land availability.

2) Protected rates of sea level are likely to be far greater than already experienced, so the record thus far may not be very indicative of future change, in which there may be non-linear responses. To direct this, it requires a sentence that compares the sum of past sea-level rise with future project rates, and which acknowledges that the past may not be a perfect guide to future responses. This also suggests qualifying the conclusions, which tend to say 'for the next century', when changes over that timescale seem too uncertain to be so confident about (e.g. line 152).

We agree with this comment and have made a number of adjustments to the manuscript and supplementary information to appropriately deal with this issue:

- i) We have included the sea level data from Funafuti atoll, the only site with a sea level gauge within the archipelago, to provide a framework for discussion on this point (see new supplementary Figure 3). We make this point and refer to the new figure in lines 80-82 and 146-148).
- ii) The sea level data shows there has been a 3.9 ± 0.4 mm/yr increase in sea level over the timeframe of analysis (1977-present) which corresponds to the period of analysis of island change (40 years). Of note, this rate of rise equates to

approximately 0.4m/100 yrs, which is comparable to the lower-mid range of projections for the next century. We have inserted a sentence outlining this recent rapid rate of sea level change in the archipelago at the end of the discussion.

iii) We now discuss our findings in light of future changes in sea level in lines 247-255. We note that:

'Significantly, our results show that islands can persist on reefs under rates of sea level rise on the order of 3.9 ± 0.4 mm/yr over the past four decades (Supplementary Figure 3) equating to an approximate total rise of 0.15 m. This rate is commensurate with projected rates of sea level rise across the next century under the RCP2.6 scenario mid-point rate of 4.4 mm/yr (range 2.8-6.1 mm/yr)⁴⁸. However, under the RCP8.5 the projected rate of sea level rise will double to 7.4 mm/yr (range 5.2-9.8 mm/yr). Under these higher sea level projections it is unclear whether islands will continue to maintain their size, though the dynamic adjustments observed are expected to occur at faster rates placing a premium on establishing ongoing monitoring of island morphological dynamics.'

3) *The causes of the observed changes are scarcely considered, yet surely these matter a lot. How much of the accretion is due to construction? what role do storms play? Given most beaches are constructed by forams, what is known about biological productivity now and into the future? and so on... An additional paragraph that draws on any available evidence of causes for Tuvalu or related systems seems necessary and might also help inform thinking about the drivers of change and the scope for adaptation.*

We agree and have now inserted a substantive section in the discussion (new lines 155-217) to examine key environmental processes that promote change and the likely influence of these process in changing islands within the archipelago. The section now considers, sea level, wave regime, storms and sediment supply in accounting for island change. We do note that most beaches are not composed of foraminifera as suggested by the reviewer but rather coral and coralline algae.

Minor corrections (some of which pick up on the above).

Abstract:

- ***line 6, consider changing to 'are key drivers expected to render'...***

Changed. Though the abstract has now been rewritten to conform to the Nature Communications format.

- ***line 8, consider changing 'physical change' with 'changes in shorelines'***

Changed to 'shoreline change'.

- ***line 9, consider adding a phrase that explains the method (e.g. 'using remote sensing data')***

Change instituted 'Using remotely sensed data...'. Abstract now modified to conform to the Nature Communications format.

- ***line 10, change to 'local sea level'***

Instituted change.

- **line 14, is a bit awkward, could be 'challenge conventional constructions of erosion and inundation and key drivers of migration'.**

Yes – but we feel the results have much deeper implications for approaches to adaptation that include possibilities other than migration. We have inserted 'erosion and loss'. We are unable to extrapolate assertions of changes to inundation from the dataset we present.

- **suggest deleting lines 15, 16, and half of 17, so it reads ...'suggesting scope for new, flexible...'**

We have reworked the abstract to comply with the Nature Communications format. The changes have taken into account the suggested deletion.

- **line 18. consider changing to 'coastal defence'**

We have instituted this suggestion.

- **line 19, reads poorly, consider rewriting.**

We have reworked the abstract to comply with the Nature Communications format and therefore, have reworked this sentence

- **Para 1. Bit long and obtuse, this could be one sentence.**

We have rewritten the first paragraph in order to clarify and simplify.

- **Lines 38-48. This is long and a bit confusing. The relationship between the science, popular narratives, what is known about adaptation, and what actually happens could be made clearer.**

We have rewritten this section in order to clarify the points made.

- **Line 66 - here is where comparison of the sum of past SLR and projected future SLR could be made.**

We have responded to the issue related to sea level and have inserted discussion on past vs future sea level change (lines 247-255).

- **Fig 1: does latitude play a role here? in terms of SST, and storm intensity for example?**

We have now included a substantive discussion component concerning the environmental drivers of change (new lines 155-217). This section deals directly with storm processes, a function of latitude.

- **line 176, change 'geopolitical discourses' to 'colonial associations'**

We understand that there are a broader set of influences at work in shaping population movement and settlement. A colonial imprint is certainly important in influencing the development of infrastructure and nodes of settlement but there are a wider set of geopolitical drivers at play. We have amended this sentence to refer to geopolitical influences rather than discourses.

- **ref 28, Connell has two 'la's.**

Typographical error has been corrected.

Reviewer #2 (Remarks to the Author):

We thank the reviewer for their insightful comments on our manuscript. We note they indicate the article contributes new, interesting and valuable information on an important issue and should be published. The reviewer makes a number of useful suggestions to improve the manuscript and we respond to these suggestions in detail below. The review comments are in bold text followed by our response.

Title – Technically it only provides data for one nation although other studies (inc. Webb and Kench, Ford, Mann and Westphal in Takuu, Duvat in the Tuamotus etc – uncited here) do indicate that changes in Tuvalu are certainly not unique. It is a generous assertion that it provides ‘new opportunities’ – ‘less pessimism’ might be more appropriate....

We have altered the title to be more specific about the data contribution and temper our ‘optimistic’ view point as suggested by the reviewer. We also note that we have restructured the discussion to separate the results from our analysis of all islands in Tuvalu from a discussion of the implications for atoll nations in general. This has allowed us to include additional island studies within the manuscript.

Line 6 Ref 3 Long before Barnett and Adger – Roy and Connell (Journal of Coastal Research, 1991) pointed this out and this should probably be acknowledged.

We agree. We are aware of this work but were constrained in the initial formatting re: number of references allowed. We have now included this work in the manuscript within the referencing allocation permitted.

Line 13-14 While it is true that the ‘conventional constructions’ of the media largely continue to point to island loss and the need for adaptation (a) academia has accepted and used Webb and Kench (2010) and a quick check of any citation analysis will reveal that; (b) assertions about the need for adaptation tend to link to cyclone threats, storms and overwash - as much as land loss (despite its symbolic significance). ...See also Line 49.

While we partly agree with this comment there is still evidence in the scientific literature that belief in island loss prevails (Dickinson, Hubbard et al., 2015; Albert et al., 2016; Barnett et al., 2017). For example, at a leading sea level conference in New York in June 2017 sea level rise was considered synonymous with island disappearance. While there have been a number of articles produced in recent years identifying that physical island loss may not occur (beginning with Webb and Kench), acceptance has largely been acknowledged by a small group of island geoscientists and island social scientists. Beyond this group acceptance of the results remains limited. As one of the primary authors of the Webb and Kench article (and subsequent articles) I have received a steady stream of criticism that the findings are wrong – despite lack of any evidence to refute our work. Secondly, our work has been critiqued as we have only examined a small subset of islands, which limits evaluation of adaptation implications to a few locations. Our intent with this article is to present analysis of an entire nation and highlight that consideration of island changes across the entire network of islands (land resources) provides a powerful platform for reconsideration of adaptation options. Our intent was also that publication in a high impact journal would begin to normalise the acceptance of island change/persistence.

Line 23 'Historic' is a vague word. What time period? Pre-contact? Pre-war?

We have clarified the timeframe in the text.

L26 What is a 'cultural staging platform'?!

We have amended this term to more specifically refer to pedestals for human occupation to reflect the anthropological focus on human patterns of settlement and resource use.

L31. Some think otherwise on threats to livelihoods. Jimmie Rodgers (cited in Connell 2013 Islands at Risk) claims it is NCDs. Others point to basic economic issues, especially in atoll states, like Tuvalu, that have little to export and livelihoods are challenging enough without climate change.

We have amended the sentence to indicate climate change is one of the greatest environmental threats to small islands. We also note that the reference used here was the Pacific Forum leaders statement.

L40. Again, long before Barnett, there were discussions of climate change, security and migration (see Connell, Climatic Change. A New Security Challenge for the Atoll States of the South Pacific, Journal of Commonwealth and Comparative Politics, 1993).

We agree and have added references Connell, J 1993. Climatic change: A new security challenge for the atoll states of the South Pacific, *The Journal of Commonwealth and Comparative Politics*. Vol 31 no 2, pp 173-192.

L45-48. This last sentence is quite unclear. To what extent is an archipelagic atoll state – heterogeneous? (see also Line 59). That is surely to be demonstrated, through, presumably, atoll morphology, unless the authors are suggesting other forms? What is meant by the 'historical imprint of colonial agendas'? The authors link this to 'entrenched land tenure systems' – is there a link – and what is the point? I will suggest below that 'entrenched land tenure systems', or some similar phrase, pose one kind of problem for the 'internal migration' that the authors suggest is/may be feasible.

We have reworked the lines to improve clarity. We agree that land tenure is and will be a significant difficulty in nations. Nevertheless, we believe it important to highlight that land tenure perhaps does need to be confronted in a setting where land resources are changing and will continue to constrain adaptation solutions. We have reflected on this point later in the manuscript (new lines 62, 297, 318).

L51. 'A richer set'? Authorial optimism?

We have reworded this phrase as:

'We argue that indeed there are a more nuanced set of options to be explored to support adaptation in atoll states.'

L52 'states' – we have corrected the spelling

L64. Where does the national population total of 10,600 come from? This seems a considerable over-estimate ... Oddly - the Wikipedia entry 'Demographics of Tuvalu- is useful background (though it is also the source of the 2012 census data). At least 2000 Tuvalu- born live in NZ and the de facto population of Tuvalu is probably around 9500.

We obtained the population data from the official statistics which were analysed by Dr Ward Freisen, (Tuvalu Central Statistics Division, 2012 Census of Population and Housing) and downloadable from the following link <http://tuvalu.poggis.spc.int/>.

We note that the 2002 census data had a resident population of approximately 9,500.

L96. Tepuka spelt Te Puka in supplementary tables.

Corrected spelling to ensure consistency.

L98. 'highlights' Corrected spelling.

L105 'yields' Corrected spelling.

L106 'provides' Corrected spelling.

L121 Again – useful to point to the 'existing narratives' of land loss. Cite? Storlazzi? Dickinson? Kelman? Albert et al?

We have significantly expanded this section in response to the other two reviewer's comments (line 155-217). This has afforded the opportunity to discuss the mechanisms responsible for the detected island change. As a consequence we have included a number of additional references, including those identified by the reviewer.

L126 Ref 25. Useful to cite Bayliss-Smith, Physical effects of hurricane Bebe upon Funafati atoll, Tuvalu, Australian Geographer, 1987.

Bayliss-Smith's 1988 article was largely based around observations in the Solomon Islands. In the revisions we have included articles describing the impact of hurricane Bebe on the atoll (Maragos et al., 1973; and Baines and McLean, 1976). We have also included the article by Fitchett – which was published in Australian Geographer.

L128 Sentence is rightly cautious about attribution of alterations to particular combinations (?) of ongoing anthropogenic and other influences on morphology but then (L130-131) goes on to predict a 'markedly different trajectory' from that which others have envisaged (Again- who are these straw men?] Where did the caution evaporate?

L128 was summarising some of the difficulties in assigning environmental drivers to the observed patterns of change in island shorelines. Of note, we have now considerably expanded this section to discuss the different drivers of change and their possible influence (lines 155-217).

Line 130-131 (now 220-223) is based on the fact that our aggregate results of island expansion differ from the still commonly asserted notion that islands will disappear. However, we further note that wider factors shaping habitability need to be considered (lines 243-246).

L151 The claim that islands will remain 'habitable' over the coming century is based on area alone. It takes no account of any possible impact of cyclones and droughts (both of which have unusually influenced Tuvalu in this decade) – let alone any aspirations that Tuvaluans might have for different kinds of lives, or global economic fluctuations.

We agree with this comment in part and have amended the sentence to be more precise that the changes will provide the physical foundation for habitation. We argue that the physical presence of land is an essential precursor for habitation. Cyclones and droughts can and do impact communities. Whether they render islands uninhabitable is questionable. In Tuvalu communities have experienced and withstood multiple cyclone and drought events in the past, which provides an analogue to suggest islands do remain habitable in the medium to long-term. Other external factors such as global economic fluctuations are certainly influences on whether populations may remain or move. However, that is factor well beyond the focus of this manuscript.

L166. 'Counter-intuitively .. reef platform islands are the least populated'. That presumes that the atolls are homogeneous (although the authors previously point to heterogeneity). The reef islands either do not have lagoons (or very small enclosed lagoons) that means both that access is often difficult but, more importantly, marine diversity is more limited. That very much limits livelihoods and reduces their potential for resettlement.

We have amended the text to clarify this point. While we understand the resource constraints that shape livelihood decisions we would suggest, alongside the reviewer's other observation that contemporary (and aspirational) lifestyles of Pacific communities can be different from historic practices. For example, Vaitupu is an island that has experienced past infrastructure investment to support educational and economic opportunities.

L174. 'Ill-matched' – in what sense?

We feel the explanation in the paragraph is consistent with the topic sentence. We identify that the greatest population concentrations are on smaller and more dynamic islands. We have amended this sentence to further clarify this point.

L178 Connell (2003) is strange citation here, even though it refers to socio-economic changes in Tuvalu. Far better to cite Connell, Vulnerable Islands: Climate Change, Tectonic Change, and Changing Livelihoods in the Western Pacific, The Contemporary Pacific (2015), which has a more detailed analysis of population shifts in the atoll states. The scale of the argument seems to shift here to the atoll states more generally, in which case it is not clear what the evidence is for Tarawa and Majuro being on smaller, less stable islands.

We have amended the reference according to the reviewer's suggestion.

L193-197. It is appropriate to re-imagine (L190) but there are three migration streams in most atoll contexts: (a) resettlement – a failure in Kiribati – either in the Line or Phoenix islands (yet that is the mode that the authors prefer) as it was in the Marshall Islands decades earlier, and re Gilbertese in Solomon Islands, and has only really worked in unusual circumstances in pre-colonial times, and then with cautionary tales-hence, with one minor exception, are no longer being contemplated, (b) urbanisation – with problematic consequences - on which there are multiple studies, (c) international migration – and Tuvalu is following the ‘Polynesian model’ – with numbers growing steadily in NZ with no growth in Tuvalu. Marshallese migration is even more dramatic. In other words Tuvaluans are shifting from the outer islands (a concept that is missing here) especially and moving into urban centres nationally or internationally. Moving to small resource-poor islands flies in the face of aspirations (even without the land tenure considerations). Connell reviews these resettlement problems in Australian Geographer (2012) and with particular reference to the failures of Carteret Islands resettlement in Asia-Pacific Viewpoint (2016)

By now since my cover has probably been blown, I had the same positive thoughts about the potential for outer island atoll development three decades ago (Pacific Studies, 1986) but the reluctance of people to move to resource-poor islands, without kinship ties and thus access to land, and the inability of government to develop – or even maintain – infrastructure for these islands has proved me wrong. What Wilmsen (sp) and Webber say about resettlement (even based on China) is entirely valid, but the converse is that the ‘recipients’ of this planning have little interest in developing plans. The first sentence of the next paragraph (L198-199) is thus entirely true ..but it is not just at national scale.

We largely agree with the above comments. Is the time right with heightened concern to re-examine this possibility? The driver of migration is frequently economic and related to livelihood and family aspirations. The resettlement to resource poor islands is certainly problematic (even just from an economic perspective) if these relocations are not reflective of community wants, underpinned by viable infrastructure that is well supported and the tyranny of geography (the ability to move efficiently and cost effectively between islands) is redressed. These commentaries are outside the current scope of our work but we would encourage a re-engagement with what alternative pathways may look like. While acknowledging the cultural complexities, not least at issue here is the requirement for significant and sustained economic investment, including the opportunities for economic growth and sustained adaptive capacity. Small scale community based adaptation projects may fail to support significant or transformative improvements in livelihoods, however the collective investment in projects in the Pacific is not insignificant. Coupled with wider philanthropic and industry interest there may be opportunities to pursue more forward looking dialogues about what at times are depicted as seemingly insurmountable barriers.

L292. Citation refers to RMI hence sentence needs some clarification.

Amended sentence to indicate following the methods adopted in the study from RMI.

Reviewer #3 (Remarks to the Author):

We thank the reviewer for their positive and constructive comments. Of note they highlight the thorough analysis and surprising results and are supportive of the manuscript being published. The reviewer suggests additional discussion and some points of clarification are warranted and we outline our response to these points below. The review comments are in bold text followed by our response.

1. Definition of island. What are the minimum dimensions of a shoal to be considered an island? What are the characteristics? Given the methods, I assume that the authors only measured vegetated islands but that should be clarified in the manuscript.

The analysis adopts the edge of vegetation to define the stable section of islands. This is clearly articulated in the methods (lines 352-357). This definition has become widely adopted in island change studies as it reflects the more stable core of an island that might change in position at timescales longer than seasonal fluctuations in beach position. In addition it is such vegetated islands that communities select to inhabit, rather than unvegetated and highly changeable sand shoals.

2. The authors discuss different types of islands and how ones are more stable than others; island types should be clearly defined. Perhaps another colour could be used for island types in Supplementary Figure 2. Atolls should be somehow distinguished from platforms in the tables and figures in the main manuscript. That would help the readers follow the discussion in the manuscript more easily.

We agree and have made the following amendments to the manuscript.

- i) We have colour coded islands that have accreted and those that have eroded in the Supplementary Figure 2.
- ii) Reef platform islands are denoted by square symbols in Figure 1 and atoll rim islands are denoted by solid circles. In addition we have placed a light blue circle around islands that have significant habitation. We have added text to the caption to highlight these differences.
- iii) We have differentiated reef platform islands in Figure 2 by placing RP next to relevant data lines.

3. The opening statement in Page 7 imply that the authors expect a continuity in the changes that they have measured; I assume that they are taking into account a constant rate of sea level rise. What would happen if rates of sea level rise accelerate? There are examples in the literature showing that some islands within The Solomon Islands underwent extreme erosion under faster sea level rise.

We have amended a discussion to reflect expected changes as a consequence of increased sea level change. This section occurs in the new lines 247-255. We now explicitly discuss the current rates of change in the context of recent sea level and consider the implications for accelerated rates of sea level change into the future against the IPCC global projections.

4. *The sediment sources are not discussed. I wonder whether bleaching due to warming would mean that there will be less sediment in the future. I do acknowledge that this is not the scope of this paper but a couple of sentences acknowledging the uncertainty would soften some of the claims and open the discussion towards future steps in research of island stability.*

As noted by the reviewer a full examination of sediment supply is well beyond the scope of this article and few studies have attempted to examine this important issue. However, we have now inserted a discussion of the potential drivers of island change. In this discussion we consider the role of sediment delivery to islands, particularly that driven by storms, and its importance in influence observed changes in islands (new lines 155-194).

5. *Still on the sediments, is there a relation between the island sediment composition and its stability or tendency to erode or accrete?*

There is no clear relationship between island sediment composition and stability. We do see that some islands in the NW of two atolls have eroded and a number of these are composed primarily of sand. However, this is not consistent and we have a number of gravel islands that have also reduced in size.

6. *What about other effects of climate change like, for example, changes in storm frequency, direction, and intensity? The studied period has seen several cyclones in the area (e.g., Bebe) that caused dramatic changes in the area.*

We agree. We have now included a discussion of the effect of multiple environmental processes in driving observed changes in islands. This includes, wave processes, sea level, storms and sediment supply (new lines 155-217). We make specific mention of the cyclone Bebe and its subsequent impact on island change that occurred over the ensuing decades. We note that projections of changes in storm frequency and intensity are very unclear for this sector of the Pacific making such discussion very speculative as the impacts on islands are not direct and involve spatial differences depending on reef condition, which controls sediment supply, and consequently might yield contrasting geomorphic outcomes. In summary, we have included a more full discussion of environmental processes and explored the nuances of how they can influence island change based on our observations.

7. *What about the wave/wind climate in the area? Is the study period representative of a wide range of conditions including climatic cycles that might affect wave and wind direction? I assume this is the case, given the length of the study period, however the authors should state it. Further analyses could be made considering the degree of exposure to wave energy of the erosional vs accretional transects.*

The revised article now directly addresses the wave regime in the archipelago and potential shifts in wave direction (lines 155-181). Of note, studies have shown there has been no appreciable adjustment in either the direction or magnitude of waves since 1979 in this region (Trenham et al., 2014; Bosserelle et al., 2015).

8. *It is not clear whether the erosion that has happened has affected essential infrastructure for the islands' inhabitants. I feel that this should be stated somewhere in the paper as the islanders might not care whether the island is growing in one side if they are losing their infrastructure on the other.*

We are able to report that patterns of change have had no direct impact on 'essential infrastructure'. Indeed, construction of essential infrastructure arguably may have influenced island change. We have added a section to the discussion in which we explore anthropogenic actions in the context of our island change data (new lines 195-209). We have also added a sentence indicating patterns of change have not directly impacted essential infrastructure to this point.

9. *Why is that platform islands are more resilient? And why is that they are less populated? Is there a socio-economic reason (e.g., less resources, water...)?*

This is a good question and one that is stimulating our ongoing research programme. Our working assumption is that wave processes (refraction) trap sediments on reef platform surfaces rather than potentially driving them across reef platforms to fill lagoons (as is the case in atolls, or reef with substantive lagoons). We have now alluded to this in the discussion, but feel we lack definitive evidence to substantiate this claim at this time.

With respect to population density we do explore the political and socio-economic factors for such differences. However, arguably the water resources on such islands are better than atoll rim islands, though lagoonal resources are absent on reef platform islands. The spatial pattern of population distribution likely reflects historical decisions (including those factors identified above). However, contemporary communities live in a different context (some are less reliant on natural resources) and this can alter considerations of habitation.

10. *Supplementary figure 2 helps a great deal when understanding the discussion of the paper. I think that aside from stating the types of islands, the figure should show which are the inhabited islands and which islands hold the most important cities/infrastructure. Equally, this figure should show the spatial distribution of the accreting, stable, and eroding islands by using different colours or markers. This addition will help understanding the discussion about the socio-economic and cultural costs of relocating the nation's inhabitants. Marking all inhabited islands might be difficult or next to impossible to do for all the studied islands given the scale of the figure but pointing out the main cities should be achievable.*

We agree with the reviewer's suggestion and have made the following amendments to Supplementary Figure 2.

- i) We have colour coded islands that have accreted from those that have eroded.
- ii) We have identified inhabited islands and their populations by inserting circular symbols with population values within them, which are pinned to the relevant island. We believe that the population values are diagnostic of major population centres.

11. *Finally, how representative is Tuvalu of the possible evolution of other atoll nations?*

It is a little unclear what this comment refers to. In terms of evolution the Tuvaluan archipelago is representative of developmental modes observed in other atoll settings. Furthermore, our results are consistent with studies of other atoll islands, though these

have only presented a very small sub-sample of islands within an archipelago. Consequently, on the basis of some supportive data, we believe the Tuvalu data is representative of changes taking place in other atoll nations. We have amended the latter sections of the discussion to highlight the generalities that can be extended to other atoll archipelagos.

Detailed Comments:

1. I don't think the title is clear enough, the term mosaics of habitability sounds more like a GIS tool and does not send a clear message.

We have modified the title to improve clarity.

2. Abstract:

2a. Please add "locally" to the statement about the sea level rising three times the global average.

We have changed the abstract according to the reviewer's suggestion.

2b. The results' highlights are three points, the first two seem to be the same point. Please revise punctuation in third point, it should be ";and,

We have taken the reviewer's suggestion into account in revising the entire abstract.

3. The text is clear but there is a bit of wordiness to it, for example, at the end of Page 1 the sentence "islands have been conceptualised as cultural staging platforms" is not very clear and could be worded otherwise." The same goes to the last sentence on page 6 "continual changing mosaic of physical land resources"

We have taken the reviewer's suggestions into account in reworking the sections identified by the reviewer. We believe/hope the sections are now more clear.

4. Use of word spectre on the second page; I am guessing the authors mean spectra.

No, we intended to use the word spectre, which is appropriate in the context it is written.

5. Figure 3 needs arrows indicating the lagoonward direction, it is not clear to the reader.

We agree. Arrows have been inserted.

6. Supplementary Figure 1 needs a world map or something to put everything in context.

We agree. We have included a regional scale map as panel A in Supplementary Figure 1.

7. There is a typo on the * in Supplementary Table 2, it says gavel instead of gravel. How was the island sediment composition obtained? Visually? Please state it.

We have corrected the typographical mistake.

Island sediment composition was obtained from detailed descriptions and analysis of sediments from each island published by McLean and Hosking (1992). We have included this reference at the foot of the Table.

Reviewer #3 (Remarks to the Author):

1. We have altered the manuscript title to:

Patterns of Island Change and Persistence Offer Alternate Adaptation Pathways for Atoll Nations

We believe the title needs to convey both the fact that many islands are persistent but they are also changing in size and position on reef surfaces.

2. The use of Km^2 is contained within lines 273-275. The reason for this change in units is that we wished to report island population densities as per Km^2 as is the international convention. Elsewhere we report changes in island area using Ha.
3. We have inserted 'at' twice the global average.
4. Due to word limitations in the abstract we have not included the precise IPCC concentration pathways. However, we do explicitly state this later in the manuscript where we discuss future trajectories of islands in the context of future sea level rise.
5. We have defined what we mean by small (<1 Ha), medium (1-10 Ha) and large (>10 Ha) islands. We have included these definitions where references to size appear in the text.
6. We have deleted 'to' from line 312 to improve clarity as suggested by the reviewer.
7. We have deleted 'donor' from line 320 as suggested by the reviewer.